# Evaluation of Biochemical Characteristics and Performance of the 3 Screen ICA ELISA Kit

**DOI:** 10.3390/ijms252212182

**Published:** 2024-11-13

**Authors:** Eiji Kawasaki, Yukari Takahashi, Tomoko Komeda, Megumi Sakuma

**Affiliations:** 1Diabetes, Thyroid, and Endocrine Center, Shin-Koga Hospital, Kurume 830-8577, Japan; 2Quality Assurance Section, Cosmic Corporation, Tokyo 112-0002, Japanm.inoue@cosmic-jpn.co.jp (M.S.)

**Keywords:** enzyme-linked immunosorbent assay, GAD autoantibody, IA-2 autoantibody, ZnT8 autoantibody, type 1 diabetes

## Abstract

We conducted a fundamental evaluation of the 3 Screen ICA ELISA kit, which can simultaneously measure three major anti-islet autoantibodies important in diagnosing and predicting type 1 diabetes, to assess its usefulness as a measuring reagent. In autoantibody-positive samples, the coefficient of variation for intra-assay variation ranged from 1.37% to 2.50%, inter-assay variation from 2.81% to 3.61%, and lot-to-lot variation from 2.01% to 8.61%, demonstrating good reproducibility. Additionally, interfering substances did not affect the autoantibody titers, and satisfying performance was observed in tests examining the sample freeze-thaw stability. Notably, even when the titer of GAD autoantibodies was below the cut-off value of the GAD autoantibody ELISA, the 3 Screen ICA signal was completely absorbed by recombinant GAD65 protein, indicating that the detection sensitivity for GAD autoantibody in the 3 Screen ICA ELISA is higher than that of the GAD autoantibody ELISA kit. Furthermore, in a study using IASP2020 samples from the Immunology and Diabetes Society, which aims to standardize anti-islet autoantibody assays, this kit achieved excellent results with a sensitivity of 96.0%, specificity of 100%, and accuracy of 98.57%. Measuring multiple anti-islet autoantibodies in combination is crucial for diagnosing and predicting type 1 diabetes. The ELISA kit used in this study is highly versatile and can be used in any measurement facility, making it extremely useful for routine testing.

## 1. Introduction

Anti-islet autoantibodies are humoral immune markers that are essential for the diagnosis and prediction of type 1 diabetes. Although the first anti-islet autoantibody was discovered in 1974 [1,2], molecular biology techniques came to be actively used to identify the autoantigens corresponding to islet-cell antibodies in the 1990s [3,4,5,6,7,8,9]. Consequently, autoantibodies to glutamic acid decarboxylase (GAD) [10], insulinoma-associated antigen-2 (IA-2) [11], and zinc transporter 8 (ZnT8) [12] were identified as major anti-islet autoantibodies in type 1 diabetes, resulting in high-throughput measurement methods using radioimmunoassay (RIA) and radioligand binding assays (RBA) being developed [13,14,15,16,17]. In Japan, autoantibodies to GAD and IA-2 by RIA methods started receiving coverage under national health insurance in 1996 and 2004, respectively, and have thus continued to be measured in general clinical practice.

Although the Islet Autoantibody Standardization Program (IASP) of the Immunology of Diabetes Society has demonstrated that RIA and RBA methods have superior sensitivity and specificity [18,19], these methods are limited to specific laboratories due to their requiring isotopes. As a result, the shift to non-RIA in vitro diagnostic reagents accelerated, and enzyme-linked immunosorbent assays (ELISA), which do not require isotope facilities, became mainstream. Consequently, many measuring reagents have switched from RIA to ELISA methods. ELISA-based measurement kits have also been developed for measuring autoantibodies to GAD (GADA), IA-2 (IA-2A), and ZnT8 (ZnT8A). In Japan, measurement kits were changed from “GADAb Cosmic” (RIA method) to “GADAb ELISA Cosmic” (ELISA method) in 2016 and from “IA-2Ab Cosmic” (RIA method) to “IA-2Ab ELISA Cosmic” (ELISA method) [20] in 2018.

More recently, multiplex technology that can simultaneously measure multiple autoantibodies has progressed, allowing for highly sensitive screening methods for anti-islet autoantibodies to be established, including multiplex electrochemiluminescence assay [21,22] using electrochemiluminescence and multiple antibody detection by agglutination-PCR [23,24] using PCR technology. That is where the multiplex 3 Screen ICA kit using the bridging-type ELISA, which has been proven to have excellent sensitivity and specificity in IASP, was conceived by RSR Ltd. (Cardiff, UK). We recently reported that this kit, which can simultaneously measure three anti-islet autoantibodies (GADA, IA-2A, and ZnT8A), served as a valuable screening tool for type 1 diabetes patients, potentially increasing diagnostic sensitivity alongside existing individual autoantibody tests [25]. However, in that study, we found a certain percentage of samples in which the results of individual autoantibodies and the 3 Screen ICA test were discrepant. These discrepancies may be due to the accuracy of this kit and interfering substances, which need to be evaluated before it can be used as a diagnostic tool in general clinical practice. Therefore, we evaluated the biochemical characteristics and performance of this ELISA kit to address the reasons associated with the discrepant results with individual autoantibodies.

## 2. Results

### 2.1. Precision

Intra-assay precision was determined with five replicate analyses of three anti-islet autoantibody-positive and two negative samples. To assess inter-assay precision, aliquots of these samples stored at −20 °C were analyzed over seven consecutive days using one lot of the kit. The mean titer, SD, and coefficient of variation (CV) for each sample are shown in Table 1. The index values of positive samples #1, #2, and #3 were 80.5 ± 1.20, 61.7 ± 0.85, and 56.6 ± 1.41, respectively, and the CV values ranged from 1.37% to 2.50%. The CV values in the inter-assay reproducibility test were between 2.81% and 3.61%. For the two autoantibody-negative samples, the intra-assay CV values were 3.02% and 6.81%, respectively, while the inter-assay CV values were 9.88% and 12.14%, respectively.

Lot-to-lot precision was determined with five replicate analyses using the same samples as in the intra- and inter-assay precision tests. The CV values for the three anti-islet autoantibody-positive samples conducted using three different lots ranged from 2.01% to 8.61% (Table 2). One of the autoantibody-negative samples had a higher CV (21.75%), but all lots showed negative 3 Screen ICA titers below the 6.5 index. These evaluations were conducted from May 2020 to June 2021.

### 2.2. Linearity and Detection Limit

Linearity was assessed by 10 replicate measurements of serial dilutions of sample Z with a negative control serum as illustrated in Figure 1. The measured vs. expected values showed good assay linearity (y = 6.259x + 7.204, r = 0.977, *p* < 0.0001). To determine the limit of detection, the absorbance of the reagent blank was measured 20 times, the mean ± 2SD was calculated, and the detection limit was defined as the 3 Screen ICA titer corresponding to an absorbance equal to the mean of the reagent blank value +2 SD. The detection limit thus calculated was 10.9 index. The linearity test and detection limit determination were conducted in February 2021.

### 2.3. Interference Studies

To detect the impact of potentially interfering substances on test assay performance, we analyzed various concentrations of hemoglobin, conjugated bilirubin, unconjugated bilirubin, chyle as intralipid, rheumatoid factor, and biotin on 6 autoantibody-positive samples (Positive A, B, C, D, E, and F). There was no substantial interference from hemoglobin up to 510 mg/dL, conjugated bilirubin up to 19.8 mg/dL, unconjugated bilirubin up to 20 mg/dL, chyle up to 1700 FTU, rheumatoid factor up to 500 IU/mL, or biotin up to 1200 ng/mL (Figure 2). Interference studies were conducted from June 2021 to July 2021.

### 2.4. Stability Study by Freeze-Thaw Cycle

Aliquots of the six autoantibody-positive samples and one negative control sample stored at −20 °C underwent up to five freeze-thaw cycles, with the 3 Screen ICA measured after each cycle. The index values did not vary significantly (CV value; 0.47–2.94%) throughout this process (Table 3). A stability study was conducted in September 2021.

### 2.5. Competition Study

To confirm whether the 3 Screen ICA ELISA can detect a trace amount of GAD antigen, we conducted competitive binding experiments using unlabeled recombinant human GAD65. These experiments employed rabbit serum positive for anti-human GAD antibodies, which was diluted with negative control serum (sample Z). We measured GADA in sample Z at a dilution factor of 1.0 using a bridging-type ELISA kit (RSR Ltd.) and found that the GADA titer was estimated to be 0.58 U/mL, which is significantly lower than the cut-off value (5.0 U/mL). Figure 3 illustrates the 3 Screen ICA index measured in serially diluted sample Z, both with and without preincubation with recombinant GAD65. The 3 Screen ICA signal of undiluted sample Z was higher than the calculated detection limit of the assay (i.e., 10.9 index), whereas the signal values of diluted samples continuously decreased as the dilution increased (i.e., as the sample concentration decreased); notably, all signals were completely blocked by preincubation with recombinant GAD65, indicating that the measured signals are specific. The competition study was performed in August 2024.

### 2.6. Performance of the 3 Screen ICA ELISA Kit

To evaluate the performance of the 3 Screen ICA ELISA kit, a blinded set from the IASP2020 workshop consisting of 90 control subjects and 50 samples from new-onset type 1 diabetes patients and their first-degree relatives (FDRs) was tested. All samples were assessed for 3 Screen ICA, GADA, IA-2A, and ZnT8A autoantibodies. The 3 Screen ICA demonstrated a sensitivity of 96.0%, a specificity of 100%, and an accuracy of 98.57%. In comparison, the individual assays, GADA, IA-2A, and ZnT8A achieved sensitivities, specificities, and accuracies of 90.0%, 97.8%, and 95.0%; 72.0%, 97.8%, and 88.57%; and 76.0%, 98.9%, and 90.71%, respectively. All samples from new-onset patients and FDRs that were positive for 3 Screen ICA were also positive for at least one individual autoantibody test, and vice versa. There were no cases where a sample was positive for the 3 Screen ICA but negative for the individual autoantibodies, nor were there samples with the reverse results. Additionally, two of the 50 cases (from new-onset patients and FDRs) were negative for 3 Screen ICA as well as for GADA, IA-2A, and ZnT8A.

## 3. Discussion

Type 1 diabetes is an autoimmune disease characterized by the selective destruction of pancreatic islet β-cells, leading to absolute insulin deficiency and marked hyperglycemia. Cytotoxic T cells are believed to play a major role in the destruction of these β-cells, and anti-islet autoantibodies are produced in response to autoantigens from the damaged β-cells [20].

To date, more than 10 anti-islet autoantibodies have been identified, and their usefulness in predicting and diagnosing type 1 diabetes has been extensively examined [20]. Among these, four are used clinically as the main autoantibodies: insulin autoantibodies, GADA, IA-2A, and ZnT8A. It is well-established that the number of positive anti-islet autoantibodies is important in predicting the onset of type 1 diabetes. Studies have reported that among individuals with high-risk human leukocyte antigen (HLA) genotypes for type 1 diabetes, approximately 70% develop the disease within 10 years of testing positive for multiple autoantibodies [26]. The American Diabetes Association classifies the onset of type 1 diabetes into three stages: stage 1, where blood glucose levels are normal despite the presence of multiple autoantibodies; stage 2, where β-cell destruction progresses and glucose intolerance appears; and stage 3, where further reduction in β-cell mass leads to hyperglycemia [27]. Therefore, measuring multiple autoantibodies is important for evaluating the risk of developing type 1 diabetes.

Moreover, combined measurement of multiple autoantibodies is crucial for improving diagnostic sensitivity in type 1 diabetes [20,28]. This necessity has led to the development of multiplex techniques that simultaneously measure multiple autoantibodies. In Japan, the first anti-islet autoantibody measured in general medical practice is GADA. However, IA-2A testing is also covered by health insurance but only when GADA is negative, which may delay the diagnosis of type 1 diabetes. Additionally, ZnT8A testing is currently not covered by health insurance. The 3 Screen ICA ELISA kit, which can simultaneously measure GADA, IA-2A, and ZnT8A, is expected to enhance the diagnostic sensitivity and enable earlier diagnosis of type 1 diabetes. In fact, studies of Japanese patients with type 1 diabetes have reported that the prevalence of the 3 Screen ICA is 11.3–14.2% higher in acute-onset type 1 diabetes and 1.6–1.9% higher in slowly progressive type 1 diabetes compared to GADA alone [25,29,30]. Conversely, this kit is also suitable for early negation of immune-mediated diabetes, which is important in selecting appropriate treatment strategies for non-immune and immune-mediated diabetes.

In this study, we evaluated the biochemical characteristics and performance of the 3 Screen ICA kit. We obtained excellent reproducibility, with the CVs for intra-assay variation ranging from 1.37% to 2.50%, inter-assay variation from 2.81% to 3.61%, and lot-to-lot reproducibility from 2.01% to 8.61%. Furthermore, interfering substances had no effect on the autoantibody titers, and good results were obtained in the examination of sample freeze-thaw stability. No particular problems were found in the biochemical characteristics. Additionally, at the Immunology and Diabetes Society workshop (IASP) aimed at improving the performance of immunoassays measuring autoantibodies in type 1 diabetes and enhancing concordance of results across laboratories [31,32], this kit achieved excellent results. Blinded measurements of samples from 50 type 1 diabetic patients and FDRs and 90 healthy controls demonstrated a sensitivity of 96.0%, specificity of 100%, and accuracy of 98.57%. There are, however, several limitations to this kit. To begin with, although it has been reported that the 3 Screen ICA titer can help estimate the individual positive autoantibodies [33], it detects the sum of GADA, IA-2A, and ZnT8A, making it difficult to accurately identify which individual autoantibodies are positive. Furthermore, as reported with the GAD autoantibody ELISA kit using the same bridging-type principle [34], a considerable number of false positive results may be obtained when plasma samples are used.

An issue often debated regarding the diagnosis of type 1 diabetes using this multiplex ELISA kit is whether patients who are 3 Screen ICA positive but negative for all individual autoantibodies should be diagnosed with type 1 diabetes. Recently, it was reported that in samples showing such discrepant results, the GADA titer was the only independent associated factor in multivariate analysis [33]. Furthermore, since the 3 Screen ICA titer and GADA titer showed a significant positive correlation, it was speculated that the 3 Screen ICA ELISA detects trace amounts of GADA below the cut-off value of the GADA ELISA kit. Our study seems to support this hypothesis, as we detected 3 Screen ICA signals even in diluted samples of GADA rabbit antiserum that were judged to be GADA negative with the GADA ELISA kit; additionally, these signals were completely abolished by pre-incubation with recombinant GAD65 (Figure 3). These results suggest that patients who are positive for 3 Screen ICA should be diagnosed with immune-mediated diabetes and that, unlike the identification of individuals at high risk for developing the disease or prediction of the progression of slowly progressive type 1 diabetes [26,35,36,37,38], type 1 diabetes can be diagnosed by measuring 3 Screen ICA alone, since the diagnosis can be made regardless of the particular anti-islet autoantibody or the number of positive autoantibodies.

Therefore, the ELISA kit used in this study, which is sensitive, highly versatile, and can be used in any facility, is highly useful for routine testing as an early/rapid and accurate method for diagnosing type 1 diabetes.

## 4. Materials and Methods

### 4.1. Reagents and Samples

The samples used in the basic evaluation of the 3 Screen ICA ELISA kit (RSR Ltd., Cardiff, UK) were rabbit antisera against human GAD65 (Positive #1, A, B), human IA-2 (Positive #2, C, D), and human ZnT8 (Positive #3, E, F) diluted with pooled serum from healthy adults; pooled serum from healthy individuals (Negative #1, #2); and rabbit serum positive for anti-human GAD antibody diluted with the negative control serum (sample Z). In addition, the negative control, GADA positive control, IA-2A positive control, and ZnT8A positive control provided with the kit were also used in each assay. Rabbit antisera against human GAD65, IA-2, and ZnT8 were kindly donated by RSR Ltd. (Cardiff, UK). Serum from healthy individuals was purchased from Kohjin Bio Co., Ltd. (Tokyo, Japan). Furthermore, the performance of the 3 Screen ICA ELISA kit was examined using 140 de-identified blinded samples (90 blood donors, 38 new-onset patients with type 1 diabetes, and 12 multiple anti-islet autoantibody-positive FDRs of individuals with type 1 diabetes from the IASP2020). According to the decoded data provided by the IASP committee, patients with type 1 diabetes comprised 23 males and 15 females with a median age-at-onset of 14.0 (range 8.0–47.0) years. The FDRs had a median age of 18.0 (range 10.0–53.0) years and included 5 male and 7 female individuals.

### 4.2. Principles and Procedure of the 3 Screen ICA ELISA

A schematic of the principles of the 3 Screen ICA ELISA is illustrated in Figure 4. The 3 Screen ICA ELISA kit (RSR Ltd., Cardiff, UK) is based on the bridging-type principle. Samples are added to the wells of an ELISA plate on which human recombinant GAD65 antigen (amino acids 1–585), IA-2 antigen (amino acids 604–979), and ZnT8 antigen (amino acids 275–369: a fusion of ZnT8 carrying either 325Trp or 325Arg with a linker peptide) have been immobilized beforehand. The anti-islet autoantibodies present in the sample then bind to the immobilized antigens. After washing the wells, biotinylated antigens (biotinylated human recombinant GAD65 antigen, IA-2 antigen, and ZnT8 antigen) are added to react with the anti-islet autoantibodies in the sample that are bound to each immobilized human recombinant antigen. After washing, streptavidin-conjugated peroxidase (SA-POD) is added to bind to the biotinylated antigen. After further washing, a substrate solution (tetramethylbenzidine) is added to develop color, and the absorbance is measured to detect the anti-islet autoantibodies in the sample. Autoantibody levels were expressed as an index defined as (OD of the test sample/OD of the reference preparation) × 100. A reference preparation was included in every assay. The cut-off value was a 20.0 index based on the 99th percentile of 159 healthy control subjects [25].

### 4.3. Individual Anti-Islet Autoantibody Assays

As appropriate, individual anti-islet autoantibodies (GADA, IA-2A, and ZnT8A) were measured in the same samples as the 3 Screen ICA using bridging-type ELISA kits (RSR Ltd.) with corresponding biotinylated autoantigens. The detailed methods for these autoantibodies were previously described [39]. The autoantibody titers were determined using a calibration curve constructed in the same run as the calibrators and expressed in U/mL. The cut-off values were 5.0 U/mL for GADA, 0.6 U/mL for the IA-2A, and 10 U/mL for the ZnT8A.

### 4.4. Interference Studies

The influence of hemoglobin, conjugated and unconjugated bilirubin, chyle, rheumatoid factor, and biotin was evaluated using Interference Check A Plus (Sysmex Corp., Kobe, Japan), Interference Check RF Plus (Sysmex Corp.), and D-Biotin (Nacalai Tesque Inc., Kyoto, Japan), respectively. Briefly, each stock solution of the Interference Check reagent or its blank reagent was mixed with anti-islet autoantibody-positive serum in a ratio of 1:9 to prepare high-concentration or zero-concentration samples for each substrate. Then, these two mixtures were combined in the ratios of 8:2, 6:4, 4:6, and 2:8 to prepare samples of each concentration, which were then measured with the 3 Screen ICA ELISA kit. For biotin interference tests, D-Biotin was dissolved in 0.1 mol/L sodium hydroxide solution, then diluted with pooled serum from healthy individuals to prepare various concentrations of biotin solutions. These were mixed with anti-islet autoantibody-positive serum in a 1:9 ratio and measured with the 3 Screen ICA ELISA kit.

### 4.5. Competition Study

Competitive binding experiments with unlabeled recombinant human GAD65 were conducted to examine whether the 3 Screen ICA ELISA detected a trace amount of GAD antigen. Serially diluted sample Z with the negative control serum was preincubated with unlabeled GAD65 (final concentration 3.8 × 10^−8^ mol/L, RSR Ltd.) or the same volume of phosphate-buffered saline for one hour at room temperature and measured with the 3 Screen ICA ELISA kit.

### 4.6. Statistical Analysis

Continuous data are expressed as mean ± standard deviation (SD). In the dilution linearity and competition tests, the dilution factor was defined as [volume of sample Z/(volume of sample Z + negative control serum)]. In studies analyzing the influence of interfering substances, the results of native samples were set to 100%, and the resulting relative averaged titers were plotted against the indices found for different interferent concentrations. Autoantibody linearity was analyzed using Spearman’s rank correlation test. The performance study calculated assay sensitivity and specificity as the percentage of case sera (new-onset patients plus FDRs) reported as autoantibody-positive and the percentage of blood donor sera reported as autoantibody-negative, respectively. Furthermore, assay accuracy was calculated as the percentage of the number of case samples identified as positive plus the number of controls identified as negative divided by the total number of case and control samples reported. A *p*-value < 0.05 was considered statistically significant. Statistical analysis was performed using StatView statistical software (version 5.0; SAS Institute, Cary, NC, USA).

## Figures and Tables

**Figure 1 ijms-25-12182-f001:**
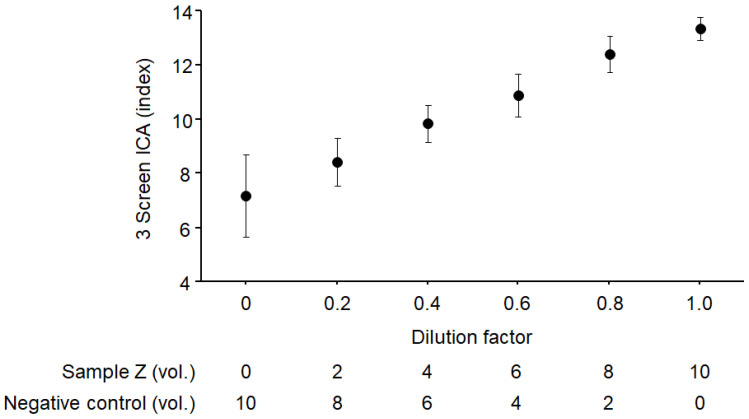
Dilution linearity test. Linearity was assessed by 10 replicate measurements of serial dilutions of the sample Z diluted with the negative control serum. The regression line of observed vs. expected values was y = 6.259x + 7.204 (r = 0.977, *p* < 0.0001). Data are mean ± 2SD.

**Figure 2 ijms-25-12182-f002:**
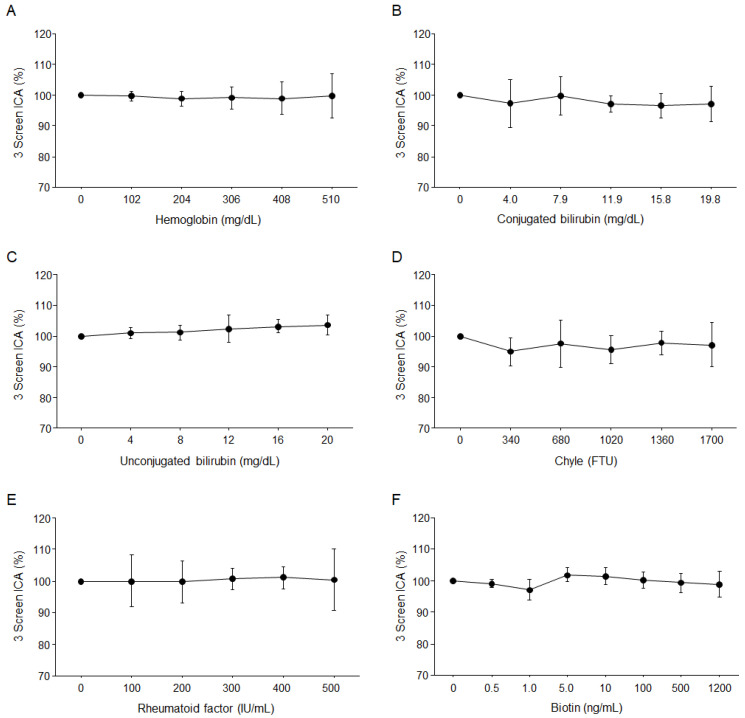
Interference of hemoglobin, bilirubin, chyle, rheumatoid factor, and biotin. (**A**). Hemoglobin; (**B**). Conjugated bilirubin; (**C**). Unconjugated bilirubin; (**D**). Chyle; (**E**). Rheumatoid factor; (**F**). Biotin. The results of samples without interferent substances were set to 100%. The data are the mean ± SD of six autoantibody-positive samples.

**Figure 3 ijms-25-12182-f003:**
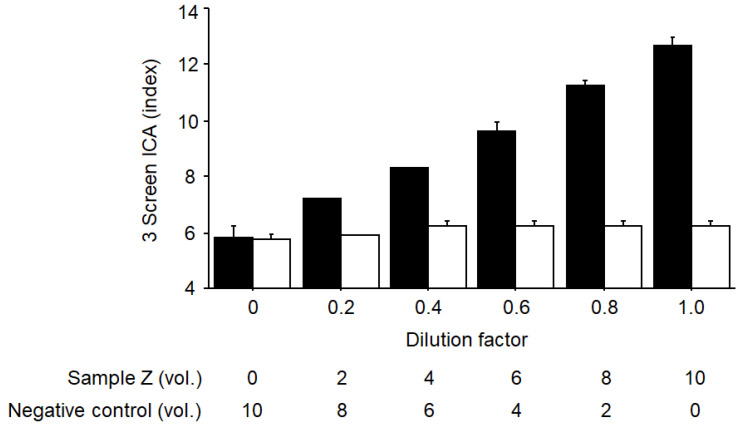
Competition study using recombinant GAD65. A 3 Screen ICA was determined in serially diluted sample Z with and without preincubation of unlabeled recombinant human GAD65 (3.8 × 10^−8^ mol/L). Data are mean ± SD of duplicate. ■, without competitor; □, with competitor.

**Figure 4 ijms-25-12182-f004:**
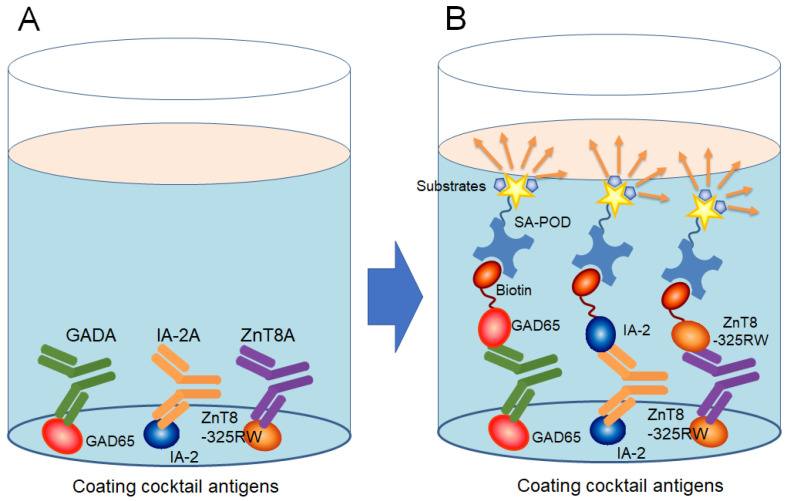
Schematic principles of the 3 Screen ICA ELISA. (**A**). Formation of complexes between solid-phase cocktail recombinant islet antigens and serum anti-islet autoantibodies; (**B**). After adding biotin-labeled antigen to the complex, the absorbance is measured at a wavelength of 450 nm. SA-POD, streptavidin-conjugated peroxidase.

**Table 1 ijms-25-12182-t001:** Intra-assay and inter-assay precision of the 3 Screen ICA ELISA.

Samples	Intra-Assay (n = 5)			Inter-Assay (n = 7)		
	Mean Titer (Index)	SD (Index)	CV (%)	Mean Titer (Index)	SD (Index)	CV (%)
Positive #1	80.5	1.20	1.49	75.1	2.11	2.81
Positive #2	61.7	0.85	1.37	59.4	2.14	3.61
Positive #3	56.6	1.41	2.50	59.5	2.09	3.51
Negative #1	6.1	0.41	6.81	4.6	0.45	9.88
Negative #2	4.0	0.12	3.02	4.3	0.52	12.14

SD, standard deviation; CV, coefficient of variation.

**Table 2 ijms-25-12182-t002:** Lot-to-lot variation of the 3 Screen ICA ELISA.

Samples	n	Lot Number and Mean Titer (Index)			Mean Titer (Index)	SD (Index)	CV (%)
		KZGIE44A	KZGIE46	KZGIE47			
Positive #1	5	74.6	71.3	80.5	75.5	4.67	6.19
Positive #2	5	58.7	52.0	61.7	57.5	4.95	8.61
Positive #3	5	58.9	57.9	56.6	57.8	1.16	2.01
Negative #1	5	4.7	4.0	6.1	4.9	1.06	21.75
Negative #2	5	4.4	4.1	4.0	4.2	0.19	4.64

SD, standard deviation; CV, coefficient of variation.

**Table 3 ijms-25-12182-t003:** Freeze-thaw cycle variation in the 3 Screen ICA ELISA.

Samples	Titer at Each Freeze-Thaw Cycle (Index)				Mean Titer (Index)	SD (Index)	CV (%)
	0	1	3	5			
Positive A	82.3	83.8	83.4	83.7	83.3	0.69	0.83
Positive B	338.3	339.4	350.6	351.0	344.8	6.92	2.01
Positive C	70.2	69.5	73.2	73.7	71.7	2.11	2.94
Positive D	375.6	384.3	377.8	397.5	383.8	9.85	2.57
Positive E	78.7	79.0	79.6	79.1	79.1	0.37	0.47
Positive F	197.5	197.7	199.2	209.6	201.0	5.78	2.88
Negative control	4.4	4.5	4.6	4.6	4.5	0.10	2.12

SD, standard deviation; CV, coefficient of variation.

## Data Availability

The datasets generated or analyzed during the current study are available from the corresponding author upon reasonable request.

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
