# Peer review of "Evaluation of Biochemical Characteristics and Performance of the 3 Screen ICA ELISA Kit"

_ijms, 2024, doi:10.3390/ijms252212182_

Round 1
Reviewer 1 Report
Comments and Suggestions for Authors
Current report performed a fundamental evaluation of the 3 Screen ICA ELISA kit and gave merits of this kit. It needs to conduct the concerns below.
1. Backgrounds of this kit must introduce in detail.
2. Rationale for current report seems obscure.
3. Source of the samples used in present evaluation was unknown.
4. Period for this evaluation must describe in detail.
5. Supplier of this kit was not indicated.
6. Limitation(s) of this assay cannot be ignored.
7. Bias of this kit is that it detects the sum of GADA, IA-2A, and ZnT8A. How to diagnose the autoantibodies in type 1 diabetes? Please describe it in detail.
8. Advantage of this kit was not mentioned in clear.
Comments on the Quality of English LanguageIt seems better to check through the professional editing service.
Author Response
Replies to the Comments from the Reviewer #1
We thank you for your evaluation of our manuscript. We have revised the manuscript according to your comments. In the revised manuscript, all changes made in the manuscript are marked using “Track changes”.
Comments and Suggestions for Authors
- Backgrounds of this kit must introduce in detail.
Response: According to your comment, we modified the sentence “Recently, we reported that the 3 Screen ICA ELISA, which can simultaneously measure three anti-islet autoantibodies (GADA, IA-2A, and ZnT8A), served as a valuable screening tool for patients with type 1 diabetes, potentially increasing diagnostic sensitivity along-side existing individual autoantibody tests.” to “That's where multiplex 3 Screen ICA kit using the bridging-type ELISA, which has been proven to have excellent sensitivity and specificity in IASP, was conceived by RSR Ltd. (Cardiff, UK). We recently reported that this kit, which can simultaneously measure three anti-islet autoantibodies (GADA, IA-2A, and ZnT8A), served a valuable screening tool for type 1 diabetes patients, potentially increasing diagnostic sensitivity alongside existing individual autoantibody tests [25].” in the Introduction section of the revised manuscript.
- Rationale for current report seems obscure.
Response: In order to clarify the rationale of this study, we added the sentence “These discrepancies may be due to the accuracy of this kit and interfering substances, which need to be evaluated before it can be used as a diagnostic tool in general clinical practice.” in the Introduction section.
- Source of the samples used in present evaluation was unknown.
Response: We added source of the samples as “Rabbit antisera against human GAD65, IA-2, and ZnT8 were kindly donated by RSR Ltd. (Cardiff, UK). Serum from healthy individuals was purchased from Kohjin Bio Co., Ltd. (Tokyo, Japan).” in the Reagents and Samples section.
- Period for this evaluation must describe in detail.
Response: Period for each evaluation was described in the Results section.
- Supplier of this kit was not indicated.
Response: Supplier of this kit is RSR Ltd. (Cardiff, UK). We added this information in the Materials and Methods section.
- Limitation(s) of this assay cannot be ignored.
Response: One of the limitations of this assay is that it detects the sum of GADA, IA-2A, and ZnT8A, which has been described in the Discussion section of the initial submitted manuscript. In addition, we added the sentence “Furthermore, as reported with the GAD autoantibody ELISA kit using the same bridging-type principle [34], a considerable number of false positive results may be obtained when plasma samples were used.” as another limitation of this kit.
- Bias of this kit is that it detects the sum of GADA, IA-2A, and ZnT8A. How to diagnose the autoantibodies in type 1 diabetes? Please describe it in detail.
Response: Unlike identifying individuals at high risk for developing or progressing type 1 diabetes, the diagnosis of type 1 diabetes is not limited to the positivity of particular autoantibodies or the number of positive autoantibodies, so type 1 diabetes can be diagnosed by detecting the sum of GADA, IA-2A, and ZnT8A. We added the sentence “Unlike identification of individuals at high risk for developing or progressing the disease, type 1 diabetes can be diagnosed by measuring 3 Screen ICA alone, since the diagnosis can be made regardless of the particular anti-islet autoantibody or the number of positive autoantibodies.” in the Discussion section.
- Advantage of this kit was not mentioned in clear.
Response: Advantage of the 3 Screen ICA ELISA kit is expected to increase the diagnostic sensitivity and enable earlier diagnosis of type 1 diabetes. Conversely, this kit is also suitable for early negation of immune-mediated diabetes, which is important in selecting appropriate treatment strategies for non-immune and immune-mediated diabetes. We mentioned this point in the Discussion section.
Comments on the Quality of English Language
It seems better to check through the professional editing service.
Response: This manuscript was edited by a native English speaker (Andrew Hamilton at Jet Communications Inc., Kurume, Fukuoka, Japan) before initial submission.

Reviewer 2 Report
Comments and Suggestions for Authors
Comments and Suggestions
Lines 14-17, “Notably, the 3 Screen ICA signal was completely absorbed by recombinant GAD65 protein even in cases where the GAD autoantibody was negative, indicating that the detection sensitivity for GAD autoantibody is higher than that of the GAD autoantibody ELISA kit”. This phrase is difficult to follow/understand.
Line 77, “Intra-assay and inter-assay accuracy of 3 Screen ICA ELISA”: it may be better to replace “accuracy” with “reproducibility” or “precision”.
Lines 79-80, Table 2: What do ‘KZGIE44A”, ‘KZGIE46”, ‘KZGIE47” stand for?
Lines 82-83, “Linearity was assessed by 10 replicate measurements of serial dilutions of sample Z (from undiluted up to a dilution of 1:5 with negative control serum)”: 1st comment: is sample Z identical with sample Positive 1# (shown in Table1/Table 2)? or with sample positive A / sample positive B (shown in Table 3)? Or?? 2nd comment: Please, provide the whole series of dilutions of sample Z along with the corresponding “dilution ratio” values.
Figure 1, Figure 3: 1st comment: What does “Dilution ratio 0” mean? please, define; 2nd comment: Is “Dilution ratio 0.2” equivalent to the highest, 1:5, dilution? 3rd comment: Which one of the columns corresponds to the undiluted sample Z??
Lines 95-101 & Lines 113-122. Please, briefly describe the exact ELISA protocols used in “interference studies” and “competition study”. Have you performed any type of “competition study” in the presence of biotin?
Line 118, “which is significantly lower than the cut-off value (5 U/mL)”: What is the corresponding detection limit of this assay?
Lines 204-208, “Therefore, developing methods for measuring anti-islet autoantibodies that can distinguish individual autoantibodies is desirable. In conclusion, the ELISA kit used in this study is highly useful for routine testing as a diagnostic method for type 1 diabetes, as it is highly versatile and can be used in any facility”: At least in my opinion, the two statements are controversial and the second (concluding) statement cannot be supported by the first one.
Author Response
Replies to the Comments from the Reviewer #2
We thank you for your evaluation of our manuscript. We have revised the manuscript according to your comments. In the revised manuscript, all changes made in the manuscript are marked using “Track changes”.
Comments and Suggestions for Authors
Lines 14-17, “Notably, the 3 Screen ICA signal was completely absorbed by recombinant GAD65 protein even in cases where the GAD autoantibody was negative, indicating that the detection sensitivity for GAD autoantibody is higher than that of the GAD autoantibody ELISA kit”. This phrase is difficult to follow/understand.
Response: We rephrased this sentence as “Notably, even when the GAD autoantibody titer was below the cut-off value of the GAD autoantibody ELISA, the 3 Screen ICA signal was completely absorbed by recombinant GAD65 protein, indicating that the detection sensitivity of GAD autoantibody in the 3 Screen ICA ELISA is higher than that of the GAD autoantibody ELISA kit.”.
Line 77, “Intra-assay and inter-assay accuracy of 3 Screen ICA ELISA”: it may be better to replace “accuracy” with “reproducibility” or “precision”.
Response: We replaced “accuracy” with “precision”.
Lines 79-80, Table 2: What do ‘KZGIE44A”, ‘KZGIE46”, ‘KZGIE47” stand for?
Response: These are lot number of the 3 Screen ICA ELISA kit purchased from RSR Ltd. (Cardiff, UK), as shown in the first row of Table 2.
Lines 82-83, “Linearity was assessed by 10 replicate measurements of serial dilutions of sample Z (from undiluted up to a dilution of 1:5 with negative control serum)”:
1st comment: is sample Z identical with sample Positive 1# (shown in Table1/Table 2)? or with sample positive A / sample positive B (shown in Table 3)? Or??
Response: Sample Z is a different sample than Positive #1, A and B.
2nd comment: Please, provide the whole series of dilutions of sample Z along with the corresponding “dilution ratio” values.
Response: We replaced the word “dilution ratio” in Figure 1 and Figure 3 with “dilution factor” in the revised manuscript. The whole series of dilutions of sample Z is shown below. We provided this information in Figure 1 and Figure 3.
|
Dilution factor |
0 |
0.2 |
0.4 |
0.6 |
0.8 |
1.0 |
|
|
Volume ratio |
Sample Z |
0 |
2 |
4 |
6 |
8 |
10 |
|
Negative control |
10 |
8 |
6 |
4 |
2 |
0 |
|
Figure 1, Figure 3:
1st comment: What does “Dilution ratio 0” mean? please, define;
2nd comment: Is “Dilution ratio 0.2” equivalent to the highest, 1:5, dilution?
3rd comment: Which one of the columns corresponds to the undiluted sample Z??
Response: In the Statistical Analysis section, we added the sentence “In the dilution linearity and competition tests, the dilution factor was defined as [volume of sample Z/(volume of sample Z + negative control serum)].”. “Dilution factor 0” means negative control serum without sample Z. “Dilution factor 0.2” means that sample Z: negative control serum = 1:4. The column with a dilution factor of 1.0 corresponds to the undiluted sample Z. In order to make it easier to understand the dilution factors, we added the entire dilution series of sample Z in Figures 1 and Figure 3 in the revised manuscript.
Lines 95-101 & Lines 113-122.
Please, briefly describe the exact ELISA protocols used in “interference studies” and “competition study”. Have you performed any type of “competition study” in the presence of biotin?
Response: We briefly described the exact ELISA protocols used in “interference studies” and “competition study” in the revised manuscript. In this study, the competitive study involves preincubation of serum with recombinant GAD65 protein to inhibit binding of the autoantigen immobilized on the 3 Screen ICA ELISA plate. The biotinylated autoantigen is added after washing the wells, so that the biotinylated autoantigen does not affect the autoantibody signal.
Line 118, “which is significantly lower than the cut-off value (5 U/mL)”: What is the corresponding detection limit of this assay?
Response: The detection limit of GAD autoantibody ELISA kit is 0.2 U/mL.
Lines 204-208, “Therefore, developing methods for measuring anti-islet autoantibodies that can distinguish individual autoantibodies is desirable. In conclusion, the ELISA kit used in this study is highly useful for routine testing as a diagnostic method for type 1 diabetes, as it is highly versatile and can be used in any facility”: At least in my opinion, the two statements are controversial and the second (concluding) statement cannot be supported by the first one.
Response: We deleted the sentence “Therefore, developing methods for measuring anti-islet autoantibodies that can distinguish individual autoantibodies is desirable.”, and modified the second statement as “Therefore, the ELISA kit used in this study is highly useful for routine testing as a diagnostic method for type 1 diabetes, as it is highly versatile and can be used in any facility.”

Round 2
Reviewer 1 Report
Comments and Suggestions for Authors
It has been revised in a good way.
Author Response
Replies to the Comments from the Reviewer #1
We thank you for your re-evaluation of our manuscript. We are glad that all comments given from you have been resolved.

Reviewer 2 Report
Comments and Suggestions for Authors
Some critical issues have been clarified in the revised manuscript. Moreover, the overall goal of the study and how the present investigation aims to shed light on previous findings of the authors (some of which have been published in International Journal of Molecular Sciences) are explained in a clearer way in the revised version. However, there is still need for further clarifications/improvement, before the manuscript can be accepted for publication.
Specific comments and suggestions are listed below:
Lines 156-159, “We measured GADA in this sample using a bridging-type ELISA kit (RSR Ltd.) and found that the GADA titer was estimated to be 0.58 U/mL, which is significantly lower than the cut-off value (5.0 U/mL)”: Sorry, this is still not clear to me, please explain: Were all positive rabbit samples (i.e., not only sample Z) also analyzed in parallel with both, the 3 Screen ICA ELISA kit and the corresponding single-autoantibody ELISA kit? (By the way, is there any difference between the “bridging-type ELISA kit”, line 157, and the “bivalent ELISA kits”, line 321??). Were serially diluted samples (sample Z, perhaps other positive samples) analyzed also with both ELISA kits, as above described?? If yes, it might be interesting to show the results obtained, e.g., in a separate Table; if no, please explain why just the experiments presented in the manuscript were designed and performed.
Lines 159-162, “Figure 3 illustrates the 3 Screen ICA index measured in serial dilutions of sample Z, both with and without preincubation with recombinant GAD65. Notably, the 3 Screen ICA signals increased as the dilution decreased (i.e. as the sample concentration increased) but were completely blocked by preincubation with recombinant GAD65”. At least in my opinion, a clearer expression would be, e.g., “Figure 3 illustrates the 3 Screen ICA index measured in serial dilutions of sample Z, both with and without preincubation with recombinant GAD65. The 3 Screen ICA signal of the undiluted sample Z was higher than the calculated detection limit of the assay (i.e., 10.9 index, line 115), while the signal values of diluted samples decreased continuously as the dilution increased (i.e. as the sample concentration decreased); notably, all signals were completely blocked by preincubation with recombinant GAD65, which is an indication that the signals measured are specific”. Does the meaning remain the same? Please, confirm.
Lines 244-249: “It was speculated that 3 Screen ICA ELISA detects trace amounts of GADA below the cut-off value of the GADA ELISA-kit. Our study supports this hypothesis, as we detected 3 Screen ICA signals even in diluted samples of GADA antiserum that were judged to be GADA negative. Additionally, these signals were completely abolished by pre-incubation with recombinant GAD65”. It might be better to change into “It was speculated that 3 Screen ICA ELISA detects trace amounts of GADA below the cut-off value of the GADA ELISA-kit. Our study seems to support this hypothesis, as we detected 3 Screen ICA signals even in diluted samples of GADA rabbit antiserum that were judged to be GADA negative with the GADA ELISA-kit; additionally, these signals were completely abolished by pre-incubation with recombinant GAD65 (Figure 3)”.
Lines 249-264, “These results suggest that patients who are positive … as it is highly versatile and can be used in any facility”: I am afraid that the authors have tried to present many different concluding remarks at once and in a way that is not completely clear. This part of the Discussion should be written again, putting the emphasis on the unique characteristics of the 3 Screen ICA ELISA kit (as they have been evaluated through the study presented) and, consequently, the specific contribution of this kit to the early/rapid and accurate diagnosis of Type-1 Diabetes.
Lines 279-280, “patients with type 1 diabetes comprised 28 males and 15 females with a median…”: Shouldn’t the total number of patients be equal to 50??
Author Response
Replies to the Comments from the Reviewer #2
We thank you for your re-evaluation of our manuscript. We have revised the manuscript according to your comments. In the revised manuscript, all changes made in the manuscript are marked using “Track changes”.
Comments and Suggestions for Authors
Lines 156-159, “We measured GADA in this sample using a bridging-type ELISA kit (RSR Ltd.) and found that the GADA titer was estimated to be 0.58 U/mL, which is significantly lower than the cut-off value (5.0 U/mL)”: Sorry, this is still not clear to me, please explain: Were all positive rabbit samples (i.e., not only sample Z) also analyzed in parallel with both, the 3 Screen ICA ELISA kit and the corresponding single-autoantibody ELISA kit? (By the way, is there any difference between the “bridging-type ELISA kit”, line 157, and the “bivalent ELISA kits”, line 321??). Were serially diluted samples (sample Z, perhaps other positive samples) analyzed also with both ELISA kits, as above described?? If yes, it might be interesting to show the results obtained, e.g., in a separate Table; if no, please explain why just the experiments presented in the manuscript were designed and performed.
Response: Unfortunately, individual anti-islet autoantibodies were not measured in other positive samples. Furthermore, serially diluted sample Z was also not analyzed with the GADA ELISA kit.
There are three reasons why GADA were not measured in serially diluted sample Z. First, since sample Z was negative for GADA even at a dilution rate of 1.0, it is easy to assume that the GADA titer in the serially diluted sample would be lower than 0.58 U/mL. Second, the antigen specificity of the GADA ELISA kit has already been verified (J Diabetes Investig 2023; 14: 570–581). Third, the purpose of this competitive experiment was to examine whether the 3 Screen ICA titer in samples judged to be GADA negative by the GADA ELISA kit is absorbed by recombinant GAD65, because it was presumed, in our previous clinical investigation (Int J Mol Sci 2024; 25: 7618), that the serum of patients who are negative for all individual autoantibodies but positive for 3 Screen ICA may contain trace amounts of GADA that cannot be detected by the GADA ELISA method.
Since the terms "bridging-type ELISA kit" and "bivalent ELISA kit" are the same, we unified them to "bridging-type ELISA kit".
Lines 159-162, “Figure 3 illustrates the 3 Screen ICA index measured in serial dilutions of sample Z, both with and without preincubation with recombinant GAD65. Notably, the 3 Screen ICA signals increased as the dilution decreased (i.e. as the sample concentration increased) but were completely blocked by preincubation with recombinant GAD65”. At least in my opinion, a clearer expression would be, e.g., “Figure 3 illustrates the 3 Screen ICA index measured in serial dilutions of sample Z, both with and without preincubation with recombinant GAD65. The 3 Screen ICA signal of the undiluted sample Z was higher than the calculated detection limit of the assay (i.e., 10.9 index, line 115), while the signal values of diluted samples decreased continuously as the dilution increased (i.e. as the sample concentration decreased); notably, all signals were completely blocked by preincubation with recombinant GAD65, which is an indication that the signals measured are specific”. Does the meaning remain the same? Please, confirm.
Response: According to your suggestion, these sentences were modified as “Figure 3 illustrates the 3 Screen ICA index in serially diluted sample Z, both with and without preincubation with recombinant GAD65. The 3 Screen ICA signal of undiluted sample Z was higher than the calculated detection limit of the assay (i.e., 10.9 index), whereas the signal values ​​of diluted samples continuously decreased as the dilution increased (i.e., as the sample concentration decreased); notably, all signals were completely blocked by preincubation with recombinant GAD65, indicating that the measured signals are specific.”.
Lines 244-249: “It was speculated that 3 Screen ICA ELISA detects trace amounts of GADA below the cut-off value of the GADA ELISA-kit. Our study supports this hypothesis, as we detected 3 Screen ICA signals even in diluted samples of GADA antiserum that were judged to be GADA negative. Additionally, these signals were completely abolished by pre-incubation with recombinant GAD65”. It might be better to change into “It was speculated that 3 Screen ICA ELISA detects trace amounts of GADA below the cut-off value of the GADA ELISA-kit. Our study seems to support this hypothesis, as we detected 3 Screen ICA signals even in diluted samples of GADA rabbit antiserum that were judged to be GADA negative with the GADA ELISA-kit; additionally, these signals were completely abolished by pre-incubation with recombinant GAD65 (Figure 3)”.
Response: According to your suggestion, we modified these sentences as “It was speculated that 3 Screen ICA ELISA detects trace amounts of GADA below the cut-off value of the GADA ELISA kit. Our study seems to support this hypothesis, as we detected 3 Screen ICA signals even in diluted samples of GADA rabbit antiserum that were judged to be GADA negative with the GADA ELISA kit; additionally, these signals were completely abolished by pre-incubation with recombinant GAD65 (Figure 3).”.
Lines 249-264, “These results suggest that patients who are positive … as it is highly versatile and can be used in any facility”: I am afraid that the authors have tried to present many different concluding remarks at once and in a way that is not completely clear. This part of the Discussion should be written again, putting the emphasis on the unique characteristics of the 3 Screen ICA ELISA kit (as they have been evaluated through the study presented) and, consequently, the specific contribution of this kit to the early/rapid and accurate diagnosis of Type-1 Diabetes.
Response: Thank you for your comment. Based on the reviewer’s suggestion, we rephrased this part of the Discussion. The sentences on the limitations of this ELISA kit was moved after the discussion on IASP 2020 results. Furthermore, we modified other sentences as “These results suggest that patients who are positive for 3 Screen ICA should be diagnosed with immune-mediated diabetes and that, unlike identification of individuals at high risk for developing the disease or prediction of the progression of slowly progressive type 1 diabetes [35-39], type 1 diabetes can be diagnosed by measuring 3 Screen ICA alone, since the diagnosis can be made regardless of the particular anti-islet autoantibody or the number of positive autoantibodies. Therefore, the ELISA kit used in this study, which is sensitive, highly versatile and can be used in any facility, is highly useful for routine testing as an early/rapid and accurate method for diagnosing type 1 diabetes.”.
Lines 279-280, “patients with type 1 diabetes comprised 28 males and 15 females with a median…”: Shouldn’t the total number of patients be equal to 50??
Response: As there was a mistake in the description of patients with type 1 diabetes and FDRs, it has been corrected as follows.
“According to the decoded data provided by the IASP committee, patients with type 1 diabetes comprised 23 males and 15 females with a median age-at-onset of 14.0 (range 8.0-47.0) years. The FDRs had a median age of 18.0 (range 10.0-53.0) years and included 5 male and 7 female individuals.”.

Round 3
Reviewer 2 Report
Comments and Suggestions for Authors
Thanks for 1st round & 2nd round responses. There are no other comments from my side.